# Reinforcing a Thermoplastic Starch/Poly(butylene adipate-co-terephthalate) Composite Foam with Polyethylene Glycol under Supercritical Carbon Dioxide

**DOI:** 10.3390/polym15010129

**Published:** 2022-12-28

**Authors:** Chih-Jen Chang, Jayashree Chandrasekar, Chia-Jung Cho, Manikandan Venkatesan, Pin-Shu Huang, Ching-Wei Yang, Hsin-Ta Wang, Chang-Ming Wong, Chi-Ching Kuo

**Affiliations:** 1Institute of Organic and Polymeric Materials, Research and Development Center of Smart Textile Technology, National Taipei University of Technology, Taipei 10608, Taiwan; 2Institute of Biotechnology and Chemical Engineering, I-Shou University, Kaohsiung 84001, Taiwan; 3CoreTech System Co., Ltd., Hsinchu 30265, Taiwan

**Keywords:** starch, polyethylene glycol, poly(butylene adipate-co-terephthalate), supercritical CO_2_, foam

## Abstract

Biodegradable foams are a potential substitute for most fossil-fuel-derived polymer foams currently used in the cushion furniture-making industry. Thermoplastic starch (TPS) and poly(butylene adipate-co-terephthalate) (PBAT) are biodegradable polymers, although their poor compatibility does not support the foam-forming process. In this study, we investigated the effect of polyethylene glycol (PEG) with or without silane A (SA) on the foam density, cell structure and tensile properties of TPS/PBAT blends. The challenges in foam forming were explored through various temperature and pressure values under supercritical carbon dioxide (CO_2_) conditions. The obtained experimental results indicate that PEG and SA act as a plasticizer and compatibilizer, respectively. The 50% (TPS with SA + PEG)/50% PBAT blends generally produce foams that have a lower foam density and better cell structure than those of 50% (TPS with PEG)/50% PBAT blends. The tensile property of each 50% (TPS with SA + PEG)/50% PBAT foam is generally better than that of each 50% (TPS with PEG)/50% PBAT foam.

## 1. Introduction

Recently, conventional plastics prepared by the petrochemical industry have been utilized for many everyday products, such as multifunctional sensors [1,2], optoelectronic devices [3], wearable electronic devices [4], etc. However, petrochemical plastics are difficult to degrade and inconvenient to collect and store for recycling. Developing biopolymeric materials as an alternative to these materials can reduce the use of non-biodegradable polymers [5], which has become one of the world’s major goals. In particular, the key advantages of biodegradable polymer-based foams are their compatibility [6], biodegradability, and renewability [7].

Starch, which can be obtained with ease, is a cost-effective biodegradable polymer. It consists of a large number of glucose components and two different structures, i.e., amylose and amylopectin. Amylose is a much smaller molecule than amylopectin, but amylose molecules in starch are more numerous than amylopectin molecules. The granular structures of starch are observed in nature and are very difficult to process directly by extrusion or injection due to the very large molecular weight. TPS consists of starch granules mixed with plasticizers, such as water or glycerin, to reduce the interaction among starch granules, and it can further be processed to obtain products. Biodegradable plastics for daily products can be an good alternative to address the difficult degradation of regular plastics, causing environmental problems. The foam products obtained from biodegradable plastics for packing and other applications have received much attention in the industrial world.

Many works associated with starch-based foams [7,8,9,10,11,12,13,14,15,16,17,18,19] include TPS foams produced by starch with high amylose content [7,10,12,13,15,16,17,19] or by grafting poly(methyl acrylate) on starch or grafting polystyrene on starch [18]. The performance of starch loose-fill foams was also investigated [11]. PBAT is a soft, biodegradable, and hydrophobic resin, but it is expensive. However, when mixed with other cheap materials, such as TPS, the PBAT/TPS blends may offer promise for many applications. TPS is a hydrophilic material. When TPS is mixed with PBAT to form blends, modification of either TPS or PBAT is required to enhance the compatibility between the two materials. Malleated TPS with PBAT [20,21,22] and TPS with malleated PBAT [15,19] are used to produce a film or foam.

Polyethylene glycol (PEG) is a water-soluble polyether compound having a nontoxic, biocompatible, and biodegradable nature. The molecular weight of PEG can lead to a liquid or a solid form, which can contribute to the melting temperature of PEG. There are plenty of applications for PEG, such as lubricants for industrial uses, excipients in pharmaceutical products, dispersants in toothpastes, antifoaming agents in food or industrial processes, binders, etc. Poly(lactic acid) (PLA) is a biodegradable thermoplastic material with excellent optical properties and high tensile strength [23,24,25]. Although the rigidity and low ductility of PLA limit its applications, when mixed with PEG, it acts as a plasticizer and has received much attention towards current research [26,27,28,29,30,31,32,33,34].

It is found that PEG can lower the glass transition temperature and the temperature of crystallization of PLA [26,27,28,29,30,31,32,33,34]. In addition, it can accelerate the crystallization process and increase the crystallinity of PLA [30,34], but it does not cause any significant change in the melt temperature of PLA [26,29,30,31,32,33,34]. The tensile modulus, tensile strength at break [26,27,28,30,31,32,33,34], flexural strength [27], and Izod impact strength of PLA/PEG are lower in comparison with PLA. However, the elongation at break among the tensile properties [26,27,28,30,31,32,33,34] either increases or decreases for PLA/PEG blends depending on the molecular weight of PEG and the amount of PEG in the PLA/PEG blend. Supercritical CO_2_ foaming of PLA/PEG blends with a higher amount of PEG achieved more open cells in the foam [34]. The foams of polystyrene/PEG blends show a bimodal cellular structure, with large and small cells coexisting, and a large cell embracing a PEG particle [35]. However, very little attention has been paid to (TPS with PEG)/PBAT blends and their foams in the literature.

In this study, we used an inexpensive industrial starch that was thermally plasticized into thermoplastic starch (TPS), acting as the main foaming material. Further, the problem of the insufficient structural melt strength of TPS was solved by the addition of biodegradable PEG or PEG with a compatibilizer. The elasticity and buffering properties of the PEG/TPS/biodegradable polyester composite were improved by expanding the molecular chains and generating intermolecular entanglements with each other. Energy saving, reduced carbon emissions, and recycling are key advantages of the presented work due to the bioplastic foaming material used. Thus, this work provides an eco-friendly material and technique for a variety of applications.

## 2. Experimental Preparation

### 2.1. Materials and Procedure

Polyethylene glycol (PEG) was purchased from Echo Chemical Co., LTD., Kaohsiung, Taiwan); tapioca starch/TPS (thermoplastic starch) from the Roi Et Group, Yannawa, Thailand; poly(butylene adipate-co-terephthalate (PBAT; Ecoflex) from BASF, Lemförde, Germany; and Silane A 6040 (SA) (236.34 g/mol) from Ya-Hu-Chi Industrial Co., Zhubei, Taiwan.

SA contains trimethoxysilyl inorganic functional groups and one reactive glycidoxy organic functional group at the other end. The crystallinity and compatibility of TPS can be altered based on the molecular weight of PEG; therefore, the effect of the molecular weight (Mw: 1000, 2000, 3000, named PEG-10, PEG-20, and PEG-30, respectively) was studied. The PEG with different molecular weights was blended with TPS with a known amount of 10 PHR (per hundred resin) with respect to the TPS concentration. However, SA with a small amount (5 PHR) is inevitable to improve the compatibility of PEG-modified TPS with PBAT.

The composite blending ratios are displayed in Table 1. Sheets of TPS/PBAT, PEG-TPS/PBAT, and (SA/PEG-TPS)/PBAT blends were produced by a hot press at a temperature of 140–160 °C, as shown in Appendix A. The sheet area and thickness were fixed at 75 mm × 75 mm and 3 mm, respectively. These sheets were directly used for foaming under CO_2_ supercritical conditions. This foam preparation experiment was repeated five times with fixed parameters to achieve a high degree of accuracy in the results, and it exhibited excellent reproducibility.

### 2.2. Surface and Functional Group Modification of TPS

First, the starch was uniformly plasticized into TPS commercially with the addition of available starch raw material to a binary solvent of water/glycerin in a plastic spectrometer (Barbender MIX, Kulturstraße, Germany). Then, the chemical structures of TPS blends were remodified based on our previous studies [36]. A functional group modifier–coupling agent of SA and different PEG (Mw: 1000, 2000, 3000) was added to the foam to enhance the number of porous cells, which afforded high flexibility (Table 1). The reaction temperature was optimized in the range of 55~70 °C, and the reaction time was 30~60 min. The pelletizer was modified and pelletized to produce surface-modified PEG/TPS and SA-PEG/TPS blends. The complete reaction mechanisms are given in Appendix A.

### 2.3. The (SA-PEG/TPS)/PBAT Biodegradable Polyester Composite Mixing

Chemical structure modifying agents were prepared by blending pre-prepared TPS and pre-mixed SA with PEG-10, PEG-20, and PEG-30 in a 1:1 ratio. Then, they were further mixed with PBAT biodegradable polyester at a 50% weight ratio. The composite mixture was transferred into a plastic spectrometer blending machine. The SA-PEG/TPS and the biodegradable polyester were uniformly mixed and dispersed into the composite. The mixing machine plastic spectrometer temperature was set at 90 °C to 145 °C, and the granulation screw speed was 50~100 rpm. Similarly, a control sample of unmodified TPS was blended with PBAT (50% TPS/50% PBAT).

### 2.4. The (SA-PEG/TPS)/PBAT Composite Foam Test Piece and Supercritical Foaming Experiment

The evenly mixed (SA-PEG/TPS)/PBAT composite was placed into a hot pressing machine to foam a film. The hot pressing process took place at a temperature of 140 °C to 165 °C; the retrieved sample appeared square in shape, with an area and thickness of 75 mm × 75 mm and 3 mm, respectively.

Then, the test samples were treated with supercritical CO_2_ using foaming equipment (200-ton capacity, Tainan City, Jing Day Machinery Industrial Co., Ltd., Taiwan) to achieve highly flexible, porous foam materials. The foaming process was controlled by adopting adequate parameters, namely a foaming temperature from 80 °C to 105 °C at two different pressure conditions of 17 MPa and 23.8 MPa. The impregnation chamber was maintained at a constant foaming temperature and pressure for 60 min. This time was sufficient to ensure that the CO_2_ in the material reached a saturated state.

### 2.5. The (SA-PEG/TPS)/PBAT Composite Foam Appearance and Internal Structure SEM Analysis

A scanning electron microscope (SEM, using a Hitachi TM4000 Plus, Hitachi High-Tech Fielding Corporation, Nagano, Japan; the WD parameter was 15.6 mm, and the HV voltage was 5.0 kV) was used to examine the foam cells’ structure, the size of the bubbles, and the dispersion of TPS in the composite foam samples after supercritical foaming.

## 3. Results and Discussion

### 3.1. FTIR Study of (P/TPS) and (SP/TPS)

The functional groups of PEG/TPS (P/TPS) prior to surface modification and after modification with SA are shown in Figure 1a by the FT-IR spectrum. In P/TPS, the peaks of IR at 3362, 2916, 1654, and 1381 cm^−1^ represent the stretching of O–H, asymmetric stretching of C–H, –CH_2_, C=O stretching, and –CH_2_-deformation, while the position of the peak at 1031 cm^−1^ may be due to the stretching of C–O–H in starch. There is a slight shift in the IR peaks of modified SA-PEG/TPS (SP/TPS to 3355 cm^−1^ (O–H stretching), 2896 cm^−1^ (C–H and –CH_2_ asymmetric stretching), 1643 cm^−1^ (C=O stretching), 1372 cm^−1^ (–CH_2_-deformation), and 1033 cm^−1^ (C–O–H stretching), respectively), indicating the interaction of SA with the TPS functional groups. The interaction is further confirmed by the shift in the peak position from 1030 to 1115 cm^−1^. The corresponding peak position of the SA modifier was observed at a low frequency. This frequency of the SA modifier is close to the peaks in P/TPS, where C–H reflects the –CH_2_– asymmetric expansion and C–O–C expansion and contraction.

H NMR Investigation of (P/TPS) and (SP/TPS).

Figure 1b shows the NMR spectrum of PEG/TPS (P/TPS). DMSO-d_6_ was used as a solvent. Compared to the literature data [37], small signals appear at 3.62 (H-6a, H-6b, and H-5), 4.6 (H-2), 5.08 (H-4), and 5.41–5.52 (H-1 and H-3) ppm, indicating the starch. The small signals at 2.49 and 3.33 ppm correspond to the –OH (hydrogen bond and water) in DMSO and starch, respectively, and the signal from the terminal proton of the –CH_2_–OH group appears at 4.44 ppm.

The NMR of SP/TPS is revealed in Figure 1b, where SA-PEG/TPS is dissolved in the DMSO-d_6_ solvent. The starch resonance is represented by a small signal at 3.6 (H-6a, H-6b, and H-5), 4.6 (H-2), 5.08 (H-4), and 5.41–5.52 (H-1 and H-3) ppm [37]. The NMR peak at 2.49 corresponds to the –OH group of DMSO and the 3.38 ppm peak is related to that of –OH (hydrogen bond and water) in starch. The terminal functional group bearing a proton resonates at 4.40 and 4.47 ppm, and the SA modifier proton of the –Si–OCH_3_ functional group shows a signal at 3.49 ppm.

### 3.2. The Foam Density of (P/TPS)/PBAT

#### 3.2.1. The Density of Foamed PBAT

In a previous study, we evaluated the density changes in PBAT foams with impregnated CO_2_ at different temperatures with two different foaming pressures [36]. The foam density of PBAT was affected proportionally by the increase in temperature under the above pressure conditions. Furthermore, the foam density obtained declined from 350 to 160 Kg/m^3^.

#### 3.2.2. The Composite Foam Density of (PEG/TPS)/PBAT

The effect of foaming temperatures ranging from 80 °C to 105 °C at a foam pressure of 17MPa and 23.8 MPa on the foam density of the four blends, i.e., 50% TPS/50% PBAT [N-1], 50% (TPS with 10PHR PEG-10)/50% PBAT [P-1], 50% (TPS with 10PHR PEG-20)/50% PBAT [P-2], 50% (TPS with 10PHR PEG-30)/50% PBAT [P-3], is illustrated in Figure 2. The highest foam density in the study for the four blends occurs at a temperature of 80 °C, in comparison with other foaming temperatures, under the two foaming pressures of 17 MPa (shown in Figure 2a) and 23.8 MPa (shown in Figure 2b). As the foaming temperature increases, the foam density for the four blends reduces. In essence, the foams produced at the foaming pressure of 23.8 MPa for the four blends have a lower foam density than those produced at the foaming pressure of 17 MPa under the same foam temperature. The lowest foam density under the foaming pressure of 17 MPa for the four blends is obtained around the foaming temperature of 100 °C; however, for a foaming pressure of 23.8 MPa, it is observed around the foaming temperature of 95 °C. Generally, the lowest foam density values of the three 50% (TPS with10PHR PEG)/50% PBAT blends are very close to one another under the foaming conditions. The foam density of the four blends rises again when the foaming temperature increases. The enhancement in the foam density indicates that the foam structure becomes weak and the shrinkage of the foam occurs at high temperatures.

#### 3.2.3. The Foam Density of (SA-PEG/TPS)/PBAT Composite

Figure 2c,d depict the foam density of the [N-1] blend and three 50% (TPS with 5PHR SA and 5PHR PEG-10, PEG-20, or PEG-30)/50%PBAT blends, i.e., [SP-1], [SP-2], and [SP-3], at six foaming temperatures and the foaming pressure of 17 MPa and 23.8 MPa, respectively. It is observed that the foam density of the [N-1] blend is higher than that of the three 50% (TPS with 5PHR SA and 5PHR PEG)/50% PBAT blends. The [SP-1] foam has the lowest foam density among the three 50% (TPS with 5PHR SA and 5PHR PEG)/50% PBAT foams. However, the [SP-2] and [SP-3] foams have a similar foam density. The lowest foam density in the study for the four foams occurs at 100 °C when the foaming pressure is at 17 MPa; when the foaming pressure is increased to 23.8 MPa, the foaming temperature shifts to 95 °C. From this, it can be seen that the four blends generate a lower foam density at the foaming pressure of 23.8 MPa than that at the foaming pressure of 17 MPa. This phenomenon is corroborated by the SEM images.

#### 3.2.4. The Density of Composite Foams at Different Foaming Temperatures and PBAT Ratios (PEG/TPS)

Appendix A show the foams (N-1), (P-1), (P-2), and (P-3), respectively. These foams are produced at various foaming temperatures ranging from 80 °C to 105 °C, at a foam pressure of 17 MPa. A good appearance and shape of the foams appeared under the foaming temperatures of 90 °C and 95 °C.

Appendix A show the foams (N-1), (P-1), (P-2), and (P-3), respectively, and the foams were produced at various foaming temperatures at a fixed foam pressure of 23.8 MPa. The sizes of the foams, without warpage or breakage, obtained at a foam pressure of 23.8 MPa were larger than those prepared at a foam pressure of 17 MPa under the same foaming temperatures. The foams with the (P-3) composition had a better appearance and shape than those with other compositions. A poor appearance and shape for the foams can be observed above 95 °C.

#### 3.2.5. The Effect of Foaming Temperature on the Density of SA-Modified Composite Foams with Different PBAT Ratios (SA-PEG/TPS)

Appendix A present the foams with (SP-1), (SP-2), and (SP-3), respectively, and the foams were produced at various foaming temperatures ranging from 80 °C to 105 °C and a foam pressure of 17 MPa. Similarly, the (SP-1), (SP-2), and (SP-3) foams under the same temperature variation at a foam pressure of 23.8 MPa are shown in Appendix A. The foams with (SP-1), (SP-2), and (SP-3) generally exhibit a good appearance and shape at the foaming conditions in the study, except at the foaming temperature of 105 °C and foam pressure of 23.8 MPa. The foams with the (SP-1), (SP-2), and (SP-3) compositions are much better in appearance and shape than those with (N-1), (P-1), (P-2), and (P-3) under the same foaming conditions, even at the high foaming temperatures applied in the study.

Moreover, with the TGA result, we can see that the reason is that SA is an epoxy-based modifier and its structural thermal stability is better. It can also increase the compatibility of TPS and PBAT and generate more entanglement characteristics between molecules (shown in Appendix A).

In general, a poor appearance and shape for the foams is observed at high foaming temperatures, such as 100 °C and 105 °C. When two foams are produced at the same foaming temperature, but at a different foam pressure of 17 MPa and 23.8 MPa, the foam produced at the higher foam pressure shows the worst appearance and structure.

### 3.3. External and Internal Structure of (PEG/TPS)/PBAT Composite Foam

(TPS/PBAT) and (PEG/TPS) Foam

The cell structure for the [N-1] foam obtained at 95 °C and the foaming pressure of 17 MPa is displayed in Figure 3a. In Figure 3a, many large openings and small TPS particles scattered in the foam can be seen. The magnified view of Figure 3a shows that the large cells and small cells exist simultaneously; TPS particles can be seen distinctly in the foam, with a clear boundary between TPS particles and cells. Appendix A exhibits the cell structure for the [N-1] foam produced at 95 °C and the foaming pressure of 23.8 MPa. Cells in the foam obtained at the foaming pressure of 23.8 MPa are more uniform and smaller than those obtained at the foaming pressure of 17 MPa. However, TPS particles are still separated from cells that arise from PBAT foaming and can generate large pores in the foam.

Figure 3b shows the cell structure for the [P-1] foam produced at the temperature of 95 °C and pressure of 17 MPa. Numerous large (TPS/PEG-10) particles are randomly scattered in the foam observed in Figure 3b, and the size of the (TPS/PEG-10) particles in the foam is generally larger than that of the TPS particles in the [N-1] foam. A gap at the boundary between (TPS/PEG-10) particles and cells for the [P-1] foam clearly exists and is similar to that of the TPS particles and cells in the [N-1] foam. However, the cell structure for the [P-1] foam exhibits a bimodal nature in terms of the cell size. One is that (TPS/PEG-10) particles generate many large pores, for which the pore sizes can be larger than 200 μm; the other is that many small cells can be seen and the cell sizes are less than 15 μm.

Appendix A presents the cell structure for the [P-1] foam produced at the foaming temperature of 95 °C and the foaming pressure of 23.8 MPa. Many large pores and small cells are scattered in the foam produced under the above-mentioned foaming conditions; however, the mean size of cells obtained at a pressure of 23.8 MPa is smaller than that of cells obtained at 17 MPa. The cell structure is shown in Appendix A, which is similar to that shown in Figure 3b. Therefore, TPS modified by PEG with a low molecular weight, 0.3 Kg/mole, does not improve the dispersion and compatibility of TPS in PBAT.

Figure 3c is the cell structure for the [P-2] foam obtained at 95 °C and a pressure of 17 MPa. When the cell structure of the [P-2] foam is compared with that of the [P-1] foam, the [P-2] foam generally has smaller (TPS/PEG-20) particles and fewer large pores than the [P-1] foam. The result indicates that TPS with P20 can obtain a better dispersive property in PBAT than that of TPS with P10. The increase in the number of (TPS/PEG-20) particles can lead to the destruction of the bimodal structure of cells observed in Figure 3b.

Appendix A represents the cell structure for the [P-2] foam obtained at the temperature of 95 °C and pressure of 23.8 MPa. Essentially, the foam produced at the foaming pressure of 23.8 MPa has a better cell structure than that of the foam produced at the foaming pressure of 17 MPa under the same foaming temperature. A bimodal structure of the cell is observed again, probably because the [P-2] foam has smaller cells from PBAT foaming produced at the high foaming pressure of 23.8 MPa than those of the foam produced at the low foaming pressure of 17 MPa.

Cell structures for the [P-3] foam obtained at the foaming temperature of 95 °C and the two foaming pressures of 17 MPa and 23.8 MP are shown in Figure 3d and Appendix A, respectively. In comparison with the effect of the three PEG components with different MWs on the TPS/PBAT blend, it is found that the particle size of TPS modified by PEG with a high MW is more uniform and smaller than that of TPS modified by PEG with a low MW in the 50% (TPS with10PHR PEG)/50% PBAT foam. However, the aggregation of small (TPS/PEG-30) particles occurs and they resemble a large particle. The [P-3] foam has a better cell structure than the [P-1] and [P-2] foams. Generally, 50% (TPS with or without PEG)/50% PBAT foams produced at a high foaming pressure have a better cell structure than = foams produced at a low foaming pressure.

SEM images for the 50% (TPS with10PHR PEG)/50% PBAT foams reveal (TPS/PEG) particles at the boundary, without interaction with cells. PEG acts as a plasticizer in the blend. The increase in MW for PEG can lead to an enhancement in the viscosity of PEG. Therefore, TPS mixed with different types of PEG can yield different viscosity levels for (TPS/PEG), which may result in differences in dispersion for (TPS/PEG) in 50% (TPS/PEG)/50% PBAT blends.

### 3.4. External and Internal Structure of (SA-PEG/TPS)/PBAT Composite Foam

(SA-PEG/TPS)/PBAT Foams with Different (SA-PEG/TPS) Ratios

The cell structure for [SP-1] foamed at a temperature of 95 °C and the foaming pressure of 17MPa is shown in Figure 4a. It is found that (TPS/SA + PEG-10) particles become small, and large (TPS/SA + PEG-10) particles almost disappear in the foam; the boundary between (TPS/SA + PEG-10) particles and cells becomes indistinct, and (TPS/SA + PEG-10) particles seem to be embedded in the foam. In general, the cell structure of the [SP-1] foam is more uniform than that of the [P-1] foam shown in Figure 3b. Appendix A represents the cell structure for [SP-1] foamed at 95 °C and a pressure of 23.8 MPa. It is observed that the two foams shown in Figure 4 and Appendix A have a similar structure, but the foam obtained at a high foaming pressure exhibits smaller cells than the foam obtained at a low foaming pressure. Therefore, SA, ass a compatibilizer, can enhance the miscibility between (TPS/SA + PEG-10) particles and PBAT and further contribute to the good dispersion of (TPS/SA + PEG-10) particles in PBAT.

Figure 4b and Appendix A display the cell structure for [SP-2] foamed at a temperature of 95 °C and pressures of 17 MPa and 23.8 MPa, respectively. It is found that the gap at the boundary between (TPS/SA + PEG-20) particles and cells becomes clear and large. The size of (TPS/SA + PEG-20) particles and the number of large cells increase, when the cell structure of the [SP-2] foam is compared with that of the [SP-1] foam. Therefore, due to the increase in the MW of PEG, (TPS/SA + PEG-20) particles show poor dispersion in the (TPS/SA + PEG-20)/PBAT blend. However, the cell structure of [SP-2] is better than that of the [P-2] foam. Figure 4c and Appendix A, respectively, exhibit the cell structures for [SP-3] foams obtained at the foaming temperature of 95 °C and the foaming pressures of 17 MPa and 23.8 MPa. The (TPS/SA + PEG-30) particles in the foam are observed more distinctly than (TPS/SA + PEG-10) particles in the [SP-1] foam. The aggregation of (TPS/SA + PEG-30) particles resulted in an increase in the number of large pores in the foam. In particular, the [SP-3] foam obtained at the foaming pressure of 23.8 MPa shows a bimodal cell structure, where small cells and large cells coexist. Moreover, the cell structure of the [SP-3] foam is close to that of the [P-2] foam in the study. The results indicate that TPS mixed with both SA and PEG with a high MW can result in poor dispersion and miscibility for (TPS/SA + PEG) particles in the (TPS/SA + PEG)/PBAT blend.

### 3.5. Tensile Properties and Elongation at Break of TPS/PBAT and SA-Modified (SA-PEG/TPS)/PBAT Composite Foam

Table 2 shows the tensile strength and elongation at break of five foams, i.e., the [N-1], [P-1], [P-2], and [P-3] foams achieved at the foaming pressure of 17MPa(A) and the [P-1] foam obtained at the foaming pressure of 23.8 MPa(*), under the foaming temperature of 95 °C. Essentially, the three 50% (TPS with PEG)/50% PBAT foams have better tensile strength and more than twice the elongation at break compared to the 50% TPS/50% PBAT foam. Among the three 50% (TPS with PEG)/50% PBAT foams achieved at the foaming pressure of 17 MPa, the [P-3] foam shows better ultimate tensile strength and elongation at break than the [P-1] and [P-2] foams. The tensile strength of [P-1] is poor and, even at the foaming pressure of 23.8MPa, it does not show an improvement in the tensile property. These results of the tensile properties are consistent with the cell structures displayed in Figure 3b–d and Appendix A.

The ultimate tensile strength and elongation at break for the [N-1], [SP-1], [SP-2], and [SP-3] foams were achieved at the foaming pressure of 17 MPa(A), and the [SP-1] foam was also obtained at the foaming pressure of 23.8 MPa(B) under the foaming temperature of 95 °C. The tensile strength and elongation at break of the three 50% (TPS with 5PHR SA and 5PHR PEG)/50% PBAT foams are much better than those of the 50% TPS/50% PBAT foam. In addition, the tensile strength is also better than that of the 50% (TPS with 10PHR PEG)/50% PBAT foams. Distinctly, the [SP-1] foam has the best tensile strength among the three 50% (TPS with 5PHR SA and 5PHR PEG)/50% PBAT foams. However, it shows a similar elongation at break to the [SP-2] and [SP-3] foams obtained under the foaming pressure of 17 MPa. It is also observed that the [SP-1] foam produced at a foaming pressure of 23.8 MPa is slightly better in terms of tensile strength and elongation at break the [SP-1] foam produced at a foaming pressure of 17 MPa. These results for the tensile properties reflect the cell structure of the [SP-1] foam, which has small cells and also good dispersion and miscibility for (TPS/SA + PEG-10) particles with PBAT in blends, as indicated in Figure 4a and Appendix A.

## 4. Conclusions

Three different molecular weights of PEG, namely PEG-10, PEG-20, and PEG-30, as a plasticizer, and SA as a compatibilizer, are used to modify TPS. PEG-10 and PEG-30 have the lowest and the highest molecular weight, respectively, and PEG-20 has an intermediate molecular weight. Moreover, 50% TPS, 50% (TPS with 10 PHR different PEGs), or 50% (TPS with 5PHR SA and 5PHR different PEGs) are mixed with 50% PBAT by weight to form blends. These blends are foamed by supercritical CO_2_ at foaming temperatures ranging from 80 °C to 105 °C and foaming pressures of 17 MPa and 23.8 MPa. The 50% (TPS with PEG-10)/50% PBAT foam has the lowest foam density among the three 50% (TPS with PEG)/50% PBAT foams under the foaming conditions, and the 50% (TPS with SA and PEG-10)/50% PBAT foam also shows the lowest foam density among the three 50% (TPS with SA and PEG)/50% PBAT foams. The foam density of the 50% (TPS with SA and PEG)/50% PBAT foam is generally lower than that of the 50% (TPS with PEG)/50% PBAT foam. The lowest foam density for those blends obtained at a foaming pressure of 17MPa is obtained at around the foaming temperature of 100 °C. However, at a high foaming pressure of 23.8 MPa, the lowest foam density for the blends occurs at a foaming temperature of 95 °C.

The tensile property and elongation at break for the three 50% (TPS with PEG)/50% PBAT foams are ultimately better than those of the 50% TPS/50% PBAT foam. The tensile strength of the three 50% (TPS with SA + PEG)/50% PBAT foams is much better than that of the three 50% (TPS with PEG)/50% PBAT foams; however, both foams have a similar elongation at break. The cell structure and tensile property of the 50% (TPS with SA + PEG-10) and 50% PBAT foam are better in comparison with those of other foams.

## Figures and Tables

**Figure 1 polymers-15-00129-f001:**
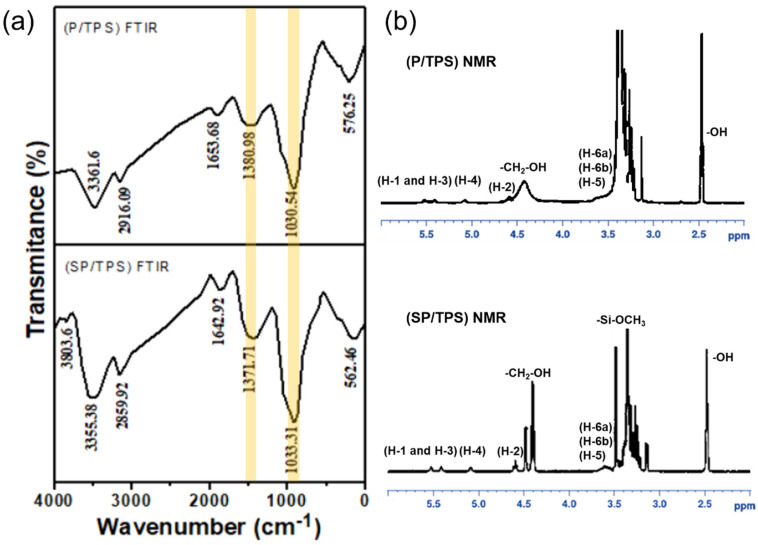
Schematic of chemical modification for analysis of TPS functional groups. (**a**) FTIR of (P/TPS) and (SP/TPS). (**b**) NMR of (P/TPS) and (SP/TPS).

**Figure 2 polymers-15-00129-f002:**
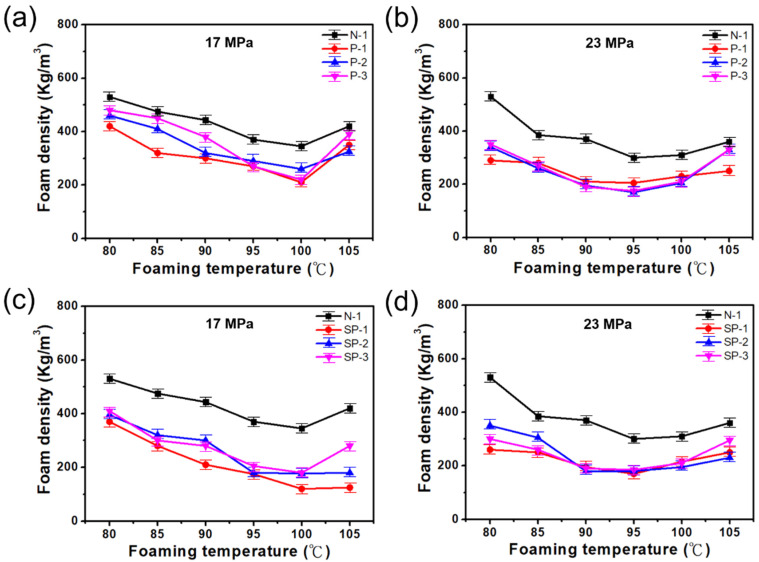
Dependence of (PEG-TPS)/PBAT and (PEG-TPS with SA)/PBAT composite foam density under six foaming temperatures and different foaming pressures. (**a**,**c**) Variation in composite foam density at 17 MPa. (**b**,**d**) Variation in composite foam density at 23 MPa.

**Figure 3 polymers-15-00129-f003:**
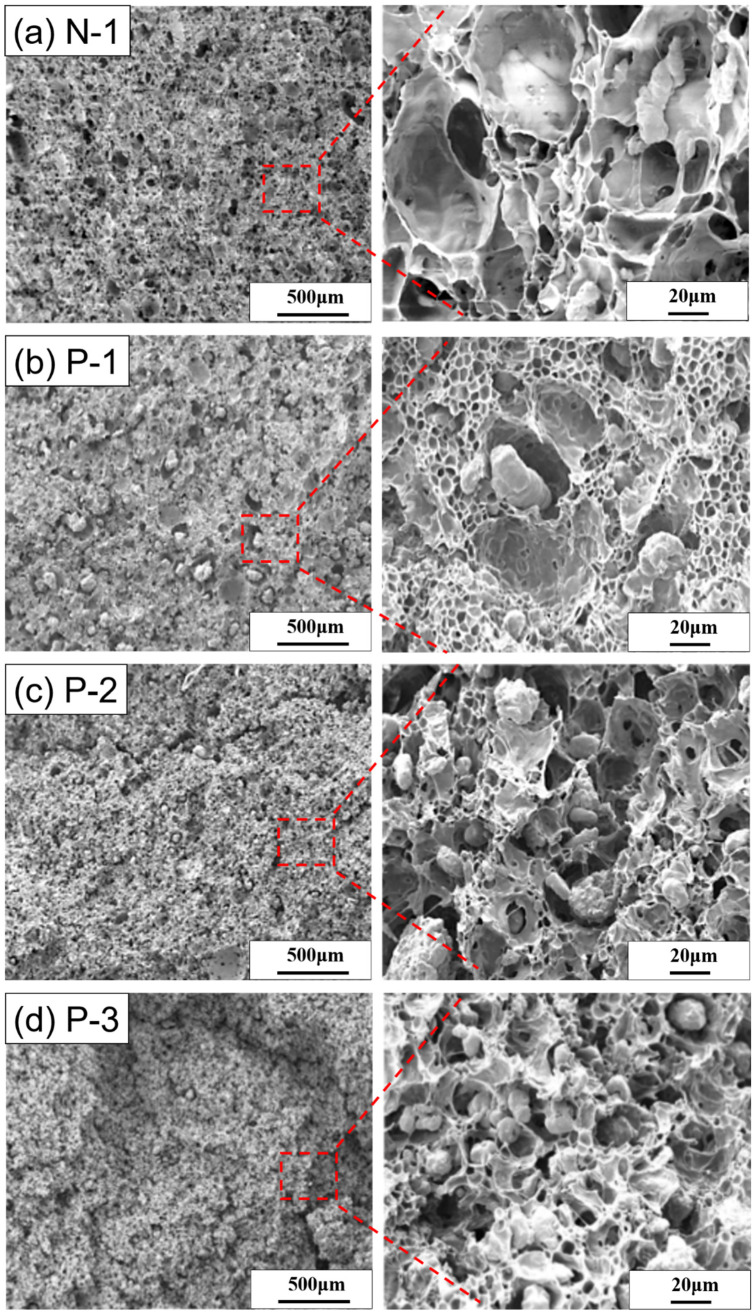
SEM images of cell structure of (**a**) [N-1], (**b**) [P-1], (**c**) [P-2], and (**d**) [P-3] at the foaming temperature of 95 °C and the foaming pressure of 17 MPa.

**Figure 4 polymers-15-00129-f004:**
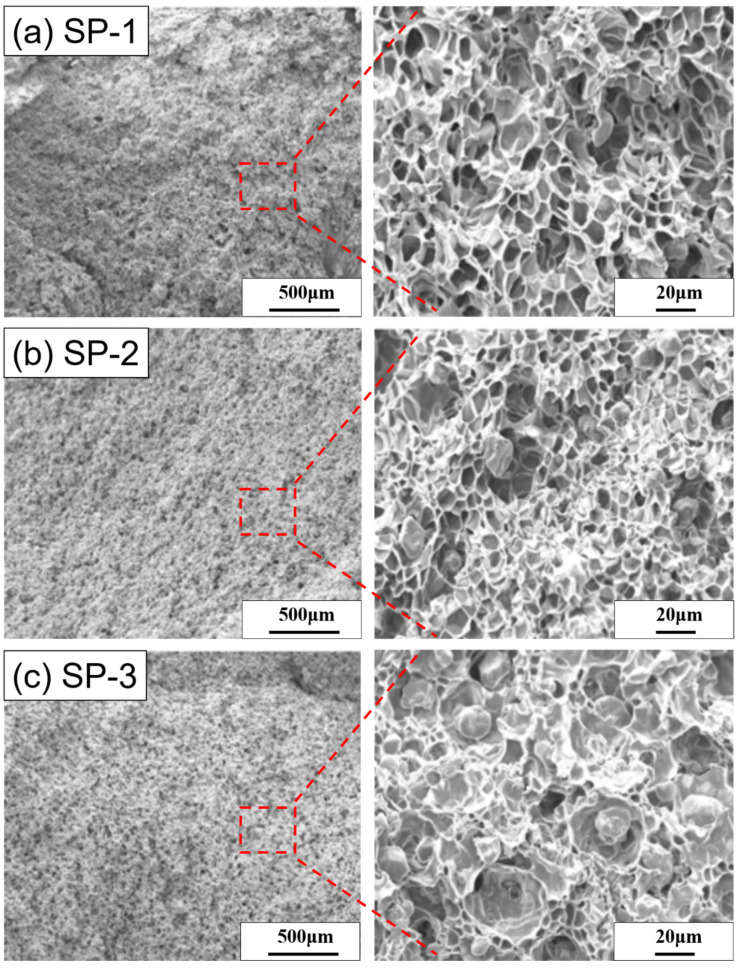
SEM images of cell structure at the foaming temperature of 95 °C and the foaming pressure of 17 MPa. (**a**) [SP-1], (**b**) [SP-2], and (**c**) [SP-3].

**Table 1 polymers-15-00129-t001:** The composite material composition ratio and corresponding symbols list.

Symbols	Blends (by Weight) Ref.
[N-1]	50% TPS/50% PBAT [36]
[P-1]	50% (TPS with 10PHR PEG-10)/50% PBAT
[P-2]	50% (TPS with 10PHR PEG-20)/50% PBAT
[P-3]	50% (TPS with 10PHR PEG-30)/50% PBAT
[SP-1]	50% (TPS with 5PHR SA + 5PHR PEG-10)/50% PBAT
[SP-2]	50% (TPS with 5PHR SA + 5PHR PEG-20)/50% PBAT
[SP-3]	50% (TPS with 5PHR SA + 5PHR PEG-30)/50% PBAT

**Table 2 polymers-15-00129-t002:** The mechanical properties of the (TPS/SA/PEG)/PBAT composite foams.

	Tensile Strength (kPa)	Elongation (%) Ref.
[N-1] (F)	226±35	20 ± 3 [36]
[P-1] * (F)	308±43	42 ± 2
[P-1] (F)	294±45	45 ± 1
[P-2] (F)	363±41	43 ± 2
[P-3] (F)	431±42	45 ± 3
[SP-1] * (F)	760±40	48 ± 1
[SP-1] (F)	745±45	45 ± 2
[SP-2] (F)	637±52	44 ± 1
[SP-3] (F)	608±44	46 ± 1

(*): The foaming temperature of 95 °C and the foaming pressure of 23.8 MPa.

## Data Availability

Not applicable.

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
