# Peer review of "Reinforcing a Thermoplastic Starch/Poly(butylene adipate-co-terephthalate) Composite Foam with Polyethylene Glycol under Supercritical Carbon Dioxide"

_polymers, 2022, doi:10.3390/polym15010129_

Round 1

Reviewer 1 Report

1. The preparation process of TPS is not clear and needs to be detailed. For example, what is the temperature of TPS processing conditions? What is the mixing time? What is the rotational speed?

 2. The preparation method of TPS/PABT PEG-TPS/PBAT, and (SA/PEG-TPS)/PBAT blends is to take out TPS after preparation and then blend with PBAT, or directly add PBAT and SA without taking out TPS? Please state the processing conditions.

 3. The PEG and SA will obviously affect their thermal properties. Please add TGA data and make statements and explanations.

 4. The foaming density of N1 is much higher than that of other samples. The author only stated the phenomenon, but did not explain the reason. What caused this? Please explain.

 5. From the mechanical properties in Table 2, is there any compatibilizer such as SA for N1 sample? Is there any data about adding SA or other compatibilizers for N1 sample? Or related literature? If yes, please add.

Author Response

根據審稿人意見逐條修改

審稿人 1:

給作者的意見和建議

  1. The preparation process of TPS is not clear and needs to be detailed. For example, what is the temperature of TPS processing conditions? What is the mixing time? What is the rotational speed?

Answer: Thanks for the reviewer's comment. We prepared the detailed information, and add in our Supporting Information. From page 3 to page 6 (Figure S1).

(A). Functional group chemically modified TPS experiment

To carry out the experiment of chemically modifying TPS with functional groups, first, commercially available starch raw material, water, and appropriate glycerin are kneaded in the kneader (Knead Machine), where the starch is uniformly thermally plasticized into TPS. Then a functional group modifying agent (coupling agent) is added to modify TPS. The functional group modifying agent includes Silane and PEG with four different molecular weights, The reaction temperature is 70-55°C and the reaction time is 30-60min .The granulator performs modification and granulation to produce modified TPS particles.

(B). Mixing experiment of modified TPS/biodegradable polyester composite

The modified TPS is added with different proportions of biodegradable polyester PBAT, and mixed with the plastic spectrometer mixing machine, so that TPS and biodegradable polyester are uniformly mixed and dispersed to form a compound. The temperature of the plastic spectrometer mixing chain equipment was set at 90~145°C, screw speed at 50~100 rpm for re-granulation.

(C). Production of modified TPS/biodegradable polyester foam test piece

The modified TPS/biodegradable polyester composite particles enter the hot press machine, the temperature of the equipment is set at 140~165°C, and the square test pieces with the size of 75mm X 75mm X 3mm are hot pressed.

(D). Supercritical foaming experiment of modified TPS/biodegradable polyester composite

Put the hot-pressed 75mm X 75mm X 3mm laminated square test piece into the mold of the supercritical foaming equipment, and obtain different foams according to different foaming parameter settings. The production parameters are as follows:

The foaming temperature is set at 80~105°C, the foaming pressure is 17 MPa and 23.8 MPa, and the CO2 impregnation time is 60 minutes. This impregnation time can make the CO2 in the material to reach a saturated state.

(E). Analysis of specific gravity, appearance and internal structure of modified TPS/biodegradable polyester foam

Observation of SEM foam structure type -The foam sample after supercritical foaming is plated with gold on the cross-section by vacuum evaporation, and then the structure of the foam cells and the size of the bubbles are observed and analyzed by SEM. Dispersion in the composite .

(F). Modified TPS/biodegradable polyester composite  for mechanical property test

The modified TPS/biodegradable polyester composite particles enter the hot press machine, the temperature of the equipment is set at 140~165°C, and the test piece is hot pressed to a size of 75mm X 75mm X 3mm. The tensile speed of the universal tensile machine is 50cm/min, and the load is 5KG for the mechanical property test.

  1. The preparation method of TPS/PABT PEG-TPS/PBAT, and (SA/PEG-TPS)/PBAT blends is to take out TPS after preparation and then blend with PBAT, or directly add PBAT and SA without taking out TPS? Please state the processing conditions.

Answer: Thanks for the reviewer's deep consideration of our manuscript. The preparation method of TPS/PABT PEG-TPS/PBAT, and (SA/PEG-TPS)/PBAT blends is directly added PBAT and SA without taking out TPS. We prepared the detailed information, and add in our Supporting Information. From page 3 to page 6 (Figure S1).

  1. The PEG and SA will obviously affect their thermal properties. Please add TGA data and make statements and explanations.

Answer: Thanks for the reviewer's deep consideration of our manuscript. We prepared the TGA data and make statements and explanations, and add in our revised manuscript (Page no.7), and Supporting Information. From page no.11 to page no.13 (Figure S7 to Figure S9).

Our revised manuscript: Besides, With the TGA result, we can find the reason is that SA is an epoxy-based modifier and its structural thermal stability is better. It can also increase the compatibility of TPS and PBAT and generate more entanglement characteristics between molecules (shown in Figure S7 to Figure S9)

Supporting Information: TGA of [N-1] sample without addition of any modifier is shown in Figure S7. The first thermal degradation occurs at the temperature of 303.57°C, the mass loss is about 11% in the pyrolysis temperature range of low-molecular glycerol, and it is biodegradable at 350°C~436°C with mass loss of polymer PBAT about 48%.

TGA of sample PEG 1000 [P-1] is shown in Figure S8. The first thermal degradation occurs at 292.79°C. Before that, the mass loss of glycerol and PEG is about 12.3%. 292-374°C is the thermal degradation temperature of TPS, 350~436℃ is the pyrolysis temperature range of biodegradable polymer PBAT, and the heat loss is about 42.7%.

TGA of sample [SP-1] with the addition of SA+PEG10 is shown in Figure S9., thermal degradation began to occur at 186°C. The thermal degradation continued upto the temperature of 209°C with weight loss of 6.6%. The second degradation occurred at the temperature of 306.42°C. It can be seen that the heat resistance of TPS[P-1] after adding SA is 14℃ higher than that of PEG10 [P-1], and it is found that the initial degradation temperature of the biodegradable polymer PBAT has increased from 350℃ to 400℃, and the final thermal degradation has been increased from 450°C to 500°C. It can be seen that the addition of SA not only increase the heat resistance temperature of TPS, but also increase the heat resistance temperature of PBAT. The reason is that SA is an epoxy-based modifier and its structural thermal stability is better. It can also increase the compatibility of TPS and PBAT and generate more entanglement characteristics between molecules.

  1. The foaming density of N1 is much higher than that of other samples. The author only stated the phenomenon, but did not explain the reason. What caused this? Please explain.

Answer: Thanks for the reviewer's deep consideration of our manuscript. The high foaming density is due to the lack of viscoelastic properties of the material and the ineffective support of the bubbles. The reason is the compatibility of TPS and PBAT. The more obvious way can be judged by the SEM image. ( shown in manuscript Figure 3. (a) [N-1] and Figure 4. (a) [SP-1] )

There are foam Figures with SA compatibilizer TPS particles it has better dispersion in PBAT and the interface between TPS particles and PBAT composite is more blurred, and the particle size is not easy to distinguish.

In the SEM of N-1 foam without adding a compatibilizer we can clearly see that the TPS particles are inside the PBAT foam, which means that the diameter of TPS particles is relatively large and PBAT is not close to the surroundings of TPS particles, so it is easy to produce larger diameter bubbles.

Basically, the bubble size distribution of this composite foam is not uniform, which leads to the phenomenon of high foam density (it can be seen that the diameter of large bubbles and small bubbles can differ by more than 10 times)

  1. From the mechanical properties in Table 2, is there any compatibilizer such as SA for N1 sample? Is there any data about adding SA or other compatibilizers for N1 sample? Or related literature? If yes, please add.

答:感謝審稿人的意見。我們沒有添加。N1樣品未添加任何相容劑或相容劑。

Reviewer 2 Report

The paper contains a very large amount of detailed experimental results  which are however conducted with systematic rather than innovation. These improvements re necessary

1.Indeed the paper has 49 references, of which 47 are in the introduction indicating that the  system studied is not very innovative.

2. Also the  results obtained are well describe but not attempt to scientific interpretation and speculation is made

3.Also  comparison with literature results is limited.

4.Line 18: PBAT is NOT a biopolymer !

5.The Conclusion section contains no new concepts but simply summarize the experimental results obtained,

Author Response

A point-by-point revision according to the reviewers' comments

Reviewer 2:

Comments and Suggestions for Authors

The paper contains a very large amount of detailed experimental results which are however conducted with systematic rather than innovation. These improvements re necessary

  1. Indeed the paper has 49 references, of which 47 are in the introduction indicating that the system studied is not very innovative.

Answer: Thanks for the reviewer's suggestion. It's very little attention has been paid to (TPS with PEG)/PBAT blend and its foam in the research. We have used different materials and different construction methods for the study.

      In this study, we used an inexpensive industrial starch that was thermally plasticized into thermoplastic starch (TPS), acting as the main foaming material. Further, the problem of insufficient structural melt strength of the TPS was solved by the addition of biodegradable PEG or PEG with compatibilizer. The elasticity and buffering property of the PEG-TPS/biodegradable polyester composite was improved by expanding the molecular chains and generating intermolecular entanglement with each other.

  1. Also the results obtained are well describe but not attempt to scientific interpretation and speculation is made

Answer: Thanks for the reviewer's deep consideration of our manuscript. We added and integrated some interpretations:

The enhancement of foam density indicates that foam structure becomes weak and shrinkage of foam happens at high temperatures.  (shown in the manuscript Page no.5)

The result indicates that TPS with P20 can perform a better dispersive property in PBAT than that of TPS with P10. The increase in the number of (TPS/PEG-20) particles can lead to the destruction of the bimodal structure of cells observed in Figure 3(b). (shown in the manuscript Page no.8)

A bimodal structure of the cell is observed again probably because [P-2] foam has smaller cells from PBAT foaming produced at the high foaming pressure of 23.8MPa than those of the foam produced at the low foaming pressure of 17MPa. (shown in the manuscript Page no.8)

SEM images for 50% (TPS with10PHR PEG)/50% PBAT foams reveal (TPS/PEG) particles at the boundary without interaction with cells. PEG acts like a plasticizer in the blend. The increase in MW for PEG can lead to the enhancement of viscosity of PEG. Therefore, TPS mixed with different PEGs can yield different viscosity for (TPS/PEG) which may result in different dispersion for (TPS/PEG) in 50% (TPS/PEG)/50% PBAT blends. (shown in the manuscript Page no.8)

Aggregation for (TPS/SA+PEG-30) particles resulted in an increase in the number of large pores in the foam. (shown in the manuscript Page no.9)

The results indicate that TPS mixed with both SA and PEG with high MW can result in poor dispersion and miscibility for (TPS/SA+PEG) particles in (TPS/SA+PEG) /PBAT blend. (shown in the manuscript Page no.9)

  1. Also comparison with literature results is limited.

Answer: Thanks for the reviewer's comment. Explanation of innovation this paper obtained patents in Europe (Germany, Britain, France), the United States, Japan, China, Taiwan, etc. in 2015. At that time, no academic papers were published. Now, all phenomena of this research are explained based on patents.

  1. Line 18: PBAT is NOT a biopolymer !

Answer: Thanks for the reviewer's suggestion. It is modified to biodegradable polymers. (shown in the manuscript Page no.9)

  1. The Conclusion section contains no new concepts but simply summarize the experimental results obtained,

Answer: Thanks for the reviewer's deep consideration of our manuscript.

Instructions on innovation this paper obtained patents in Europe (Germany, Britain, France), the United States, Japan, China, Taiwan, etc. in 2015. Now, all phenomena of this research are explained based on patents.

Citing patents to discuss

Round 2

Reviewer 1 Report

The manuscript has been improved and can be accepted for publication.

Reviewer 2 Report

All muy oints were suffciently considered .Thanks the paper is now accepatable for publication in my opinion.